# Proactive Criminal Thinking and Restrictive Deterrence: A Pathway to Future Offending and Sanction Avoidance

**DOI:** 10.3390/ijerph191811636

**Published:** 2022-09-15

**Authors:** Xin Guan, T. Wing Lo

**Affiliations:** Department of Social and Behavioural Sciences, City University of Hong Kong, Hong Kong, China

**Keywords:** crime strategy, restrictive deterrence, proactive criminal thinking, drug dealers, perceived crime benefit

## Abstract

Perceived crime benefit and criminal thinking are essential factors in predicting future offending. However, less is known about how the interaction of the two influences individuals’ perception and cognition of crime. This study explores whether proactive criminal thinking mediates the effect of perceived crime benefit, and tests whether restrictive deterrence influences these pathways. Using a drug dealer sample that was drawn from the Second RAND Inmate Survey, this paper finds that proactive criminal thinking significantly mediates the effect of perceived crime benefit on future offending, criminal self-efficacy, and future sanction avoidance. Mediation pathways are enhanced when taking a heterogeneous crime strategy as a moderator, but only in the experienced drug dealer subsample. These results suggest that proactive criminal thinking is a route for channeling the effects of perceived crime benefit, and an amplifier for bringing restrictive deterrence into play. Both roles apply to experienced offenders rather than less-experienced offenders. Integrating restrictive deterrence with individuals’ perception and cognition of crime is a meaningful attempt to fit restrictive deterrence into a broader theoretical map.

## 1. Introduction

Given that research suggests that perceived crime benefit [1] and criminal thinking [2] play essential roles in future offending, it is somewhat surprising that there is not more criminological research exploring how the interaction of the two influences individuals’ perception and cognition of crime. However, some studies have shed light on the potential role of their convergence in promoting a tendency to commit crime, including studies that are related to the dichotomous nature of criminal thinking and studies on restrictive deterrence.

### 1.1. Criminal Thinking and Future Offending

Criminal thinking is often cited as proof of the continuity of crime in criminology and related correctional work [2,3]. Criminal thinking is a distinctive thinking pattern that is used by an offender to deal with or eliminate negative feelings, including guilt and shame that are caused by their illegal activity, so that they can continue to commit crime without a psychological burden. Criminal thinking is composed of multiple mindsets, including cognitive defensive or aggressive mechanisms, such as “excusing criminal activity, justifying criminal behavior, suppressing or resisting the authorities, and denying their original malice” [4] (p. 14). The dissipation and detachment of guilt for criminal behavior is in line with moral disengagement [5,6] and neutralization techniques [7]. Both concepts consist of mechanisms/techniques that emphasize denial and ignore responsibility, which can be acquired from social learning process. Social learning theory [8] indicates that reinforcement and assimilation of deviant behaviors, attitudes, and skills in the company of deviants help to understand the formation of criminal thinking. Further, its variant, social cognitive theory [6], lays a solid theoretical basis for the cognitive and behavioral changes resulted from criminal thinking, specifically, the mechanisms that can explain attributions of deviant activity, expectations of deviant outcome, and perceptions of self-competence [9].

In Walters’ research on criminal thinking, the kernel of moral disorder is mainly taken up more by one of its subsets—positive criminal thinking, than another counterpart—reactive criminal thinking [10]. Self-report measures of moral disengagement [11] and neutralization scale [12,13] have been verified as proper proxies of proactive criminal thinking [14]. The statistical proxy relationship is made possible in part by the existence of a subscale of proactive thinking, mollification that includes items of externalizing, rationalizing, or justifying one’s criminal actions in order to relieve guilt over the consequences of these actions [10]. Both moral disengagement, neutralization, or proactive criminal thinking are nurtured in association with family members and friends who have deviant lifestyles, or with other offenders who might give feedback with deviant-reflected appraisals [15,16,17]. With the deviant-reflected appraisals, individuals tend to possess a perception of criminal identity, pro-crime attitudes, and technical and practical guidance on conducting crime. Differential association and social learning theory are often cited as theories that explain such learning processes during an individual’s interaction with deviant groups [17,18,19,20,21,22,23,24].

The pre-planned behavioral approach to crime that is highlighted within proactive criminal thinking also distinguishes it from reactive criminal thinking [10]. A pre-planned behavioral approach to crime is characterized by a pre-judgment of outcomes and an assessment of self-efficacy [25]. It is found that proactive criminal thinking predicted positive outcome expectancy for crime [26]. Hence, proactive criminal thinking is often presented in types of crime such as theft and burglary rather than violent crime that is regularly caused by the impulsive pursuit of short-term gain that is captured in reactive criminal thinking [25]. The outcome expectancy highly relies on the assessment of the capacity to commit the crime. The evaluation on crime capacity influences the feasibility of crime as a means of living in a short or long period time. This is illustrated through the sub-dimensions of entitlement and super-optimism in proactive criminal thinking: entitlement emphasizes “a misidentification of wants as needs” [10] (p. 29) that are more easily obtained by means of crime compared to other means; super-optimism addresses the belief “that one can indefinitely avoid the negative consequences of a criminal lifestyle” [10] (p. 29). In addition, crime capacity is also related to criminal self-efficacy. An individual might increase his or her perception of the positive side of crime and decrease the perceived negative outcome because of enhanced criminal self-efficacy. The potential power of criminal self-efficacy in maintaining pro-crime cognition and lifestyle is notable, as criminal self-efficacy exerts a stronger mediation effect from past crime to future crime than criminal thinking [16]. It can be glimpsed from a pre-planned behavioral approach to crime that proactive criminal thinking encompasses elements of perceived benefits and costs of crime at the juridical and economic levels. Relatedly, Walters also illustrated the economic elements in terms of physically- and mentally-facilitated crime continuity [27]. Although the author treated the economic elements as parts of low self-control, which fits better in reactive criminal thinking, this can still inspire researchers to explore perceived crime benefit in relation to proactive criminal thinking to facilitate future offending. In relation to this, it is implied in criminal thinking studies that correctional work can be achieved from intervention in criminals’ subjective views on crime benefit and cost [27].

Criminal thinking, both proactive and reactive, reflects the criminal thought process, and helps in exploring how the criminal lifestyle fosters crime [28]. Rather than there being a unidirectional route of influence between criminal thinking and criminal activity, the two interact back and forth, as people with criminal, quasi-criminal, or criminal-like lifestyles might be prone to develop a pattern of criminal thinking, and subsequently, criminal thinking results in a continued exposure to enactment of criminal behavior. One specific case study by Walters [10] illustrated how a pre-planned lifestyle facilitated proactive criminal thinking. Mitch, who carried out more non-violent offenses (four burglaries, one robbery) than violent offenses (one assault), was characterized by his family as a calculating and cunning child who planned things out before doing them. As Walters [10] (p. 233) wrote, “even his angry outbursts were premeditated.”

### 1.2. Restrictive Deterrence and Future Offending

The idea of restrictive deterrence, proposed by Gibbs [29], developed deterrence theory further by describing a dynamic process of evaluating the benefits and risks of crime when considering possible ways to avoid arrest, which can be seen as “responsiveness to detection” [1]. The “curtailment of a certain type of criminal activity by an individual during some period” [29] (p. 33) probably the most common response for avoiding sanction which was written in the definition of restrictive deterrence. The ways of avoiding sanction are complex and varied [30]. Take drug prohibition as an example. Although harsh legal punishment and vigorous police enforcement inhibit drug markets, street-level policing does not eliminate drug markets and only alters them temporarily [31,32,33]. Specifically, drug offenders’ camouflage, counter-reconnaissance, stashing products, or choosing a less severe activity are all strategies that are perceived by individual as reducing the risk [34]. Similar sanction avoidance can be seen in other types of crime, such as property crime, cyber hacking, sex crime, among others [35,36,37,38,39,40,41]. In short, drug offenders adapt to the enforcement policy and continue to commit crimes [34,42]. The introduction of restrictive deterrence fills gaps in the understanding of deterrence by exploring the question of “how” (the expression of crime) beyond the question of “whether” (the initiation and cessation of crime) [1,36]. It is crucial to understand how offenders act in these processes [39].

Similar to criminal thinking, but more directly and obviously, restrictive deterrence and its predecessor—deterrence emphasize the economic element of the crime. Deterrence theory applies the core view of rational choice theory that an offender’s decision to engage in criminal activity is a process of weighing the costs of crime against the benefits of crime, and whether or not they will be arrested is a vital part of this trade-off equation [43]. Being arrested implies a period of imprisonment, increasing the costs of crime and decreasing the benefits. Previous studies have shown support for the role of restrictive deterrence in crime continuity. Research that is based on in-depth interviews has noted that offenders care strongly about the sanction risk (i.e., the risk of arrest, prosecution, and prison), rather than committing crime recklessly, and on this basis, they give serious consideration to the possible ways to avoid punishment [1,30,38,44]. Although small in number, quantitative studies also confirm the role of restrictive deterrence. Restrictive deterrence is expected to prolong and facilitate crime continuity [45], particularly by extending the free time between two arrests [46,47], decreasing the willingness to desist from crime [48], extending the crime area [49], or being more cautious during crime [41]. It can be seen that the study of the perceptions of deterrence in criminology rests upon the responsiveness to threat sanctions [50].

As having a crime strategy is the practical result of restrictive deterrence, the impact of a crime strategy on future offending has also been progressively confirmed by empirical studies, which can be treated as supporting the theory on restrictive deterrence. If someone conducts crimes with a crime strategy rather than purely randomly or impulsively, it can be expected that he or she will obtain a higher crime income [51] and have fewer chances to desist from a criminal career [52]. What should be noted is that a crime strategy has a stronger effect than the frequency of crime (which is often controlled in the statistical model because of its robust and stable effect on criminal cognition and perception) [51]. It can be deduced that crime income relies heavily on the way in which the individual performs during the crime. Besides facilitating an income from crime, a crime strategy can also promote or enhance self-efficacy in criminal activity or a criminal career because the individual may feel skillful and talented at crime [53]. A criminal’s ability to carry out a crime may also increase their self-efficacy, especially if the crime is deliberate and involves planning, knowledge, or skill [54].

In addition, the effect of a crime strategy on the perception of crime may not be limited to the type of crime that is committed, or in other words the effect of a crime strategy may be valid across different crime types. The core idea that the capacity to commit a crime would support the successful completion of a crime is common across different crime types. As Nee and Ward found in their review, a criminal who is familiar with crime practice can assess the criminal environment automatically and efficiently, their cognitive model of the criminal environment is activated quickly, and this cognitive model of the criminal environment then guides their subsequent criminal activity behavior [55]. To be specific, they listed four detailed cognitive patterns for criminals. Chunking refers to dividing the information in the memory into chunks based on the environmental characteristics, to simplify the decision-making process. Automaticity refers to a stable memory for crime environmental cues which leads to resource-efficiency in crime commission. Situational awareness and selective preconscious attention refer to relevant cues that are aligned with a person’s expertise that are automatically noticed and given priority, rapidly anticipating the benefits and risks of the crime. Multitasking refers to processing familiar environmental cues/tasks and newly encountered situations together. This automated pattern of thought and response, or perceptual and procedural skills, helps individuals make reasonable real-time predictions [56]. The four cognitive elements, which assist the offender in rapid reaction and decision-making, can be elicited and function in different types of crime with similar surroundings. The ability to generalize knowledge to unfamiliar environments with which one has a partial acquaintance can be explained by the fact that even a small amount of information can aid and accelerate decision-making [57,58].

The impact of a crime strategy that is not constrained by crime type is also supported by studies examining spillover effects. The spillover effect in criminology, referring to the spillover outcome of the perceived crime risk, can be either the spillover within an individual across different types of crime or the spillover within a particular type of crime but across different individuals. Based on the observation by Anwar and Loughran, the updating of crime risk is not confined to the perception of a specific crime, because an arrest for an aggressive crime affects the perceptions of both violent and income crimes [59]. However, the statistical evidence does not lay a firm foundation for the spillover effect, partly because the authors set a rigorous rule and definition for “spillover.” In contrast, when looking at the spillover effect in the same crime type but across individuals, the concept of vicarious deterrence supports the spillover phenomenon with less controversy. It has been found that vicarious deterrence can hold tremendous potential in formal crime prevention initiatives [60,61], meaning that one individual’s perception of crime risk is positively associated with that of others.

Based on studies of the spillover effects of risk perception, a question could be raised regarding crime strategy. Restrictive deterrence emphasizes the perception of crime risk that is closely linked with various situations of crime strategy. The crime strategy for a particular crime type, in terms of crime perceptions and future offending, is well explored in restrictive deterrence studies. A large body of qualitative research has given us the insight that restrictive deterrence facilitates subsequent offending. While the existence of a crime strategy and its function is less likely to be questioned, issues including the spillover or quasi-spillover effect of a crime strategy are still worth discussing. To put it precisely, the question is whether the use of a particular type of crime strategy promotes another crime, or whether or not the influence of a crime strategy is crime-specific.

### 1.3. Present Study

The studies that are mentioned above are related to proactive criminal thinking and restrictive deterrence offer compelling explanations for the issue of future offending from the standpoint of perceived crime benefit. Beyond this, some details deserve further exploration. First, empirical studies on criminal thinking that emphasize economic benefits currently focus on short-term, impulsive perceived benefits, excluding the component of planned, well thought out perceived benefits. In addition, restrictive deterrence studies focus on direct effects and talk less about the impact pathway of crime strategy. This may be because restrictive deterrence is mainly studied in qualitative studies with less statistical support or evidence. One exception is a study that mentions the influence of crime strategy on crime income, however, this was not the central topic of that study [51]. Therefore, this paper attempts to explore the interactions among perceived crime benefit, proactive criminal thinking crime strategy, and future offending. 

From the perspective of theoretical investigation, the construction fits in a framework of social learning theory and social cognitive theory (Figure 1). Deviant lifestyles promote criminal thinking (phase of social learning theory), which interprets and predicts future offence and cognition on self-efficacy and capacity on crime (phase of social cognitive theory). Specifically, the theoretical model that is proposed in the current study is composed of two parts. First, inspired by Walters’ studies illustrating the causal relationship between criminal thinking and future crime, the model in the current study features mediation pathways from perceived crime benefit to proactive criminal thinking, criminal self-efficacy, future offending, and arrest avoidance. Second, to consider restrictive deterrence and take it as a practice of deviant lifestyle, crime strategy is added to the mediation model to explore its association with the other concepts within the model and, mainly, the interaction with perceived crime benefit and proactive criminal thinking. There has been little discussion on the quasi-spillover effect of crime strategy. Thus, adding a heterogeneous crime strategy to the proposed model is worth exploring.

We, therefore, present several testable hypotheses below. Specifically, we propose three hypotheses to explore the role of proactive criminal thinking in mediating the effect of perceived crime benefit, and the potential influence of restrictive deterrence (the use of crime strategies to avoid arrest/sanction) when interacting with other focal predictors:

**Hypothesis** **1a.**
*Among a sample of drug dealers, perceived crime benefit is positively associated with proactive criminal thinking, and, through such association, the chances of future offending increase.*


**Hypothesis** **1b.**
*In addition, the higher the frequency of using a crime strategy in property crime, the stronger the mediation effect from perceived crime benefit to future offending through proactive criminal thinking (Figure 2).*


**Hypothesis** **2a.**
*Among a sample of drug dealers, perceived crime benefit is positively associated with proactive criminal thinking, and, through this association, with increased criminal self-efficacy and arrest avoidance.*


**Hypothesis** **2b.**
*In addition, the higher the frequency of using a crime strategy in property crime, the stronger the mediation effect from perceived crime benefit to arrest avoidance through proactive criminal thinking and criminal self-efficacy (Figure 3).*


**Hypothesis** **3.**
*The mediating role of proactive criminal thinking in Hypothesis 1b and Hypothesis 2b is more significant among experienced drug dealers than among less-experienced drug dealers.*


## 2. Materials and Methods

### 2.1. Sample

To test the three hypotheses, this study uses data that was drawn from an extensive survey of male prisoners at twelve prisons and fourteen county jails in California, Michigan, and Texas that was conducted by Rand Corporation and administered in late 1978 and early 1979 (Second RAND Inmate Survey) [62]. The Inter-University Consortium for Political and Social Research (ICPSR) provided researchers with data that has been used to explore the criminal activities of serious offenders [63,64]. The total sample for the Second RAND Inmate Survey was 2190. The current study focuses on participants who reported their actual engagement in drug dealing during window period 3 (WP3), which refers to the period from 1 to 24 months before the offender’s current incarceration. Hence, 850 male inmates met the requirement and were included in the following analysis.

### 2.2. Measures

#### 2.2.1. Independent Variable

Perceived crime benefit is the independent variable in the current study. A total of eight self-reported items were used to gauge the participants’ perceptions of the chance of gaining a reward from carrying out the crime: *(1) having friends, (2) having money for necessities, (3) high living, (4) owning expensive things, (5) being my own man, (6) having a lot of money, (7) having a family, and (8) being happy*. Each item was rated on a 5-point Likert-type scale (no chance = 1, low chance = 2, even chance = 3, high chance = 4, certain = 5). Based on the total score, higher scores are associated with a greater chance of reaping a reward for carrying out the crime.

#### 2.2.2. Mediator Variable

Proactive criminal thinking and criminal self-efficacy are the two mediator variables in the current study. Proactive criminal thinking consisted of twelve self-reported items that appraised a participant’s cognitive attitude towards crime. Specifically, mollification (P_mo) included three items: *(1) whenever someone gets cut or shot there is usually a good reason,*
*(**2) usually someone who gets*
*cut or shot deserves it,* and *(3) because of insurance, no one is really hurt by property crime*. Entitlement (P_en) included six items: *(1)*
*one good thought about crime is the fun of beating the system; (**2) if a man only does one or two crimes a year, chances are he’ll never get caught;*
*(**3) crime is the easiest way to get what you want**; (**4) committing crime is pretty much a permanent way of life;*
*(5) men who are really good at crime never seriously think about going straight**;* and *(6) when you’ve figure it out, doing prison time is not too hard*. Super-optimism (P_so) included three items: *(**1) it is possible to get so good at crime that you’ll never get caught;*
*(2) no matter how careful you are, you won’t always get away from crime (reversed);* and *(3) if you keep doing crime, you know you will go to prison sometime (reversed)*. The items were rated on a Likert-type scale of 4 points (strongly disagree = 1, disagree = 2, agree = 3, strongly agree = 4). Based on the mean score, higher scores indicate a robust thinking pattern that lowers the negative feelings (i.e., guilt or shame) that are caused by crimes.

Criminal self-efficacy was assessed with a single item: *“Overall, in the past, how successful do you think you were in carrying out crimes?”*. The item was rated from 1 (very unsuccessful) to 4 (very successful). The drug dealer sample showed moderate confidence in their past crime experience, with a mean value of criminal self-efficacy above 2.6 and a median value of 3, indicating a perception of relative success in crime. There are potential limitations in using this single item for the dependent variable; however, as Brezina and Topalli argued, this single item can be a reliable proxy of criminal self-efficacy because of its ability to present the individual’s cognition about generalizable capability; it is thus not situation-specific [54].

#### 2.2.3. Dependent Variable

The first dependent variable, future offending, was assessed with a single item: *“What do you think the chances are that you will try to go straight when you get out [of prison]?”*. This item was rated on a 12-point scale, with values from 0 (no chance) to 11 (completely certain). The item was later reversed into point 0 presents completely certain and point 11 indicates no chance. Although this item is inadequate when compared to actual recidivism measures, previous research has used this item in models and indicated the robustness of intentions to “go straight” in predicting actual recidivism [65]. Using this future offending measure, it should be possible to make an initial exploration of the consequences of perceived crime benefit and criminal thinking. Among the drug dealer sample, around a quarter (26.94%) of the participants reported a concrete and firm intention to “go straight” or desist from crime after release. However, more than 70% of the remainder indicated that they were less than 100% sure that they intended to quit criminal activity after this incarceration.

Arrest avoidance was assessed with a single item: *“Do you think you could do the same crime(s) again without getting caught?”*. This item was rated as a dichotomy, with 1 = no and 2 = yes.

#### 2.2.4. Moderator Variable

Crime strategy was assessed with twelve items that appraised the participant’s crime process when conducting a property crime during the window period 3 (this applied to 63% of the sample): *(1) worked out a plan for the crime before you went out to do it, (2) found places or persons with a lot of money, (3) learned about alarms, hours, or money transfers, (4)*
*decided to do the crime on the spot, (5) worked out an escape plan before doing the crime, (6) got special equipment such as burglary tools, (7) worked with partners, (8) lined up a fence or buyer before the crime, (9) used tips to line places up, (10) only cased a place or person just before the crime, (11) stole a car or got a gun that could not be traced,* and *(12) followed a person to a safe place to do the crime.* Each item was rated on a 4-point Likert-type scale (never = 0, sometimes = 1, usually = 2, always = 3). A higher mean score indicates a higher frequency of using a crime strategy when conducting property crime. For those in the targeted sample who did not commit property crimes, the score for crime strategy was set to 0.

A network analysis study using big data from the Swedish National Register of Suspected Criminal Offenders found that the relationship between property crime and drug crime is very close [66]. Offenders who have committed property offenses are likely to commit drug offenses, and offenders who have committed drug offenses are likely to commit property offenses. A potential explanation for this association is that an offender’s drug use habit connects his need to deal drugs and his need to steal to gain money to support his drug use. Shoplifting and several similar acquisitive crimes are a valid path to obtaining crime income for drug use [67]. In addition, the pattern of drug use, such as the drug type and frequency of use, are also predictors of the drug-related crimes that are committed. An association between drug dealing and intense drug use, such as high-frequency injection, has been indicated by several studies [68,69].

#### 2.2.5. Control Variable

There were 7 control variables, including age (in years), race (1 = White, 0 = non-White), education, marriage, and length of WP3 (in months), perception of self as a drug dealer (0 = No, 1 = Yes), and the number of arrests for drug dealing.

### 2.3. Analytic Strategy

Descriptive analysis was conducted to describe the criminality of the drug dealer sample. Exploratory factor analysis (EFA), confirmatory factor analysis (CFA) and structural equation model (SEM) were conducted to evaluate the construct’s quality of PCB, PCT, CS and association among them. Several recommended indices were used to evaluate the goodness of model fit, including the Chi-square, df, Tucker–Lewis Index (TLI), root mean square error of approximation (RMSEA), and standardized root mean square residual (SRMR). CFI and TLI values that are close to or higher than 0.95, RMSEA and SRMR values that are close to or lower than 0.08 indicate an adequate fit of a model to the observed data [70]. Analysis was conducted in RStudio 2022.02 using {lavaan} package [71].

Mediation analysis was used to examine whether proactive criminal thinking channels the effect of perceived crime benefit on future offending, criminal self-efficacy, and arrest avoidance. Subsequently, moderated mediation analysis was conducted to test whether and in which direction crime strategy affected the mediation pathways that were described above. Within this step, we conducted the analysis three times with different samples (i.e., total sample, experienced sample, and less-experienced sample) to explore the potential random effect due to the sample characteristics. The significance of the two indirect (mediating) effects and the related moderated mediation was evaluated using biased bootstrapped 95% confidence intervals. Significance was indicated by a confidence interval that did not include zero. Covariance among measurements were computed. No covariance at or above 0.70 was found, indicating no collinearity. Multiple variables were controlled during analysis to avoid obtaining a pseudo-effect. 

## 3. Results

### 3.1. Demographic Analysis

The target sample was composed of 850 male prisoners that were aged from 14 to 60 (M = 25.45, SD = 6.64) (Table 1). The racial/ethnic breakdown of the sample was 48% White, 37% Black, and 12% Latino/Chicano, and the remaining 5% was composed of Asian (0.5%), Indian/Native American (1.1%), and other (1.9%). More than 35% of the participants had finished 10th–11th grade education, and around 20% had finished high school. A relatively high proportion (27%) of the participants had attended college or above. As for family life experience, nearly 60% had never married, and about 20% had married and were still in a marriage. The drug crime experience of the participants varied, but they all acknowledged having engaged in drug dealing during WP3. As for the continuity in drug dealing activity, around one third of the participants (31.3%) had only committed this crime in WP3, and a smaller percentage of them (25.8%) had committed drug dealing in WP3 and WP1 or WP2. The majority of the participants (42.9%) had committed drug dealing across all three window periods. Half of them (52%) thought of themselves as drug dealers, while the rest did not. The number of arrests for drug dealing ranged from 0 to 8, with a mean value of 0.489. Only 14% of the participants had been arrested for drug dealing leading to their current incarceration, implying that most had experiences of other crime types. On average, a participant had committed three offenses of different crime types, with, on average, 2.39 property crimes and 0.56 violent crimes. The length of WP3 ranged from 1 to 25 months, with a mean value of 14.43 months. 

### 3.2. Exploratory Factor Analysis (EFA) and Confirmatory Factor Analysis (CFA)

EFA was conducted to form a proactive thinking measurement (Appendix A). Parallel analysis suggests that the number of sub-measurements should be three to be in line with expectation; however, one item that was intended to put in mollification was rotated into entitlement, and two items had low factor loading (item in mollification (*“**because of insurance, no one is really hurt by property crime**”*) was removed for its low factor loading and cross factor loaded during EFA; item in entitlement (*“when you’ve figured it out, doing prison time is not too bad”)* was removed for its low factor loading during EFA). The result of EFA indicated that the items are appropriate for subsequent factor analysis and SEM with KMO = 0.72, Bartlett’s test of sphericity is significant (Chi-square(66) = 992.54, *p* < 0.001).

The CFA confirmed measurement and structure model for further analysis. The model fit indices are listed in Table 2. The structure of PCT suggested from the result of first-order CFA was basically the same as EFA with only one item in entitlement removed because of its low factor loading (item in entitlement (*“men who are really good at crime never seriously think about going straight”*) was removed for its low factor loading during the first-order CFA). The model fit of PCT’s first-order and second-order CFA was both good with Chi-square = 42.177, df = 24, CFI = 0.965, TLI = 0.947, RMSEA = 0.034, SRMR = 0.04. For PCB, the CFA verified a 7-item measurement with two pairs of error correlation among the items (item in perceived crime benefit (*“having a family”*) was removed for its low factor loading during CFA; items *“having living”* and *“owning expensive things,”* and items *“being my own man”* and *“being happy”* were the two pairs that were error correlated). Its model fit was good with Chi-square = 52.478, df = 12, CFI = 0.960, TLI = 0.930, RMSEA = 0.08, SRMR = 0.046. For CS, the CFA confirmed a 10-item measurement (items in crime strategy (*“learned about alarms, hours, or money transfers”; “decided to do the crime on the spot”*) were removed for low factor loading). It’s model fit was good with Chi-square = 133.974, df = 35, CFI = 0.955, TLI = 0.942, RMSEA = 0.078, SRMR = 0.039. Composing the above measurement models together, the CFA for structure model also reached good model fit indices. All of the covariance among the measurement models were significant, ranging from [0.193, 0.549]. Hence, the structural model that was taken into next step included PCT (consisted of sub-scale P_mo (2 items), P_en (4 items), and P_so (3 items), PCB (7 items)), CS (10 items), and three single-item factors (single-item measurement model was constructed followed instruction in Gana et al. [72] (p. 151)): FO, CSE, and AA (Appendix A).

### 3.3. Mediation and Moderation Analysis

Table 3 presents the results of two mediation analyses involving Hypothesis 1a (perceived crime benefit → proactive criminal thinking → future offending) and Hypothesis 2a (perceived crime benefit → proactive criminal thinking → criminal self-efficacy → arrest avoidance). Testing the two mediation pathways with bootstrapped 95% confidence intervals reveals the presence of significant mediation by all three variables. The mediation pathway representing Hypothesis 1a shows a significant effect with zero not included in the bootstrapped 95% confidence interval (β = 0.583, BCBCI = [0.252, 0.914]). On the other hand, the mediation pathway representing Hypothesis 2a shows a significant and positive effect with the lower and upper bound of the bootstrapped 95% confidence interval both higher than 0 (β = 0.044, BCBCI = [0.013, 0.075]). The effect size is less significant than the mediation pathway representing Hypothesis 1a. Hence, Hypothesis 1a and Hypothesis 2a are supported. The detailed regression parameter is listed in Appendix A.

Table 4 (columns 2 to 4) presents the moderated mediation analysis incorporating the crime strategy for property crime into the above mediation pathways from perceived crime benefit to proactive criminal thinking. When proactive criminal thinking is the outcome measure, the effect of crime strategy on its own, reaches a significant effect size (β = 0.061, BCBCI = [0.026, 0.097]), as well as its interaction with the perceived crime benefit is positively associated with the outcome measure (β = 0.017, BCBCI = [0.002, 0.032]). The effect size of the mediation pathway for Hypothesis 1a grows when the moderator variable crime strategy increases. The moderation effect of crime strategy on proactive criminal thinking subsequently enhances the mediation effect of perceived crime benefit on future offending (β = 0.079, BCBCI = [0.012, 0.147]), which supports Hypothesis 1b. Similarly, the moderation effect of crime strategy increases the mediation effect of the perceived crime benefit on crime self-efficacy (β = 0.03, BCBCI = [0.007, 0.052]) and arrest avoidance ((β = 0.015, BCBCI = [0.005, 0.025]), which supports Hypothesis 2b. Figure 4 and Figure 5 presents the standardized path coefficients of moderated mediation structural equation model. The detailed regression parameter is listed in Appendix A.

The last hypothesis of the current study was intended to explore the difference between the group of experienced drug dealers and the group of less-experienced drug dealers, since it is believed that these groups do not share same crime cognition. For experienced drug dealers who committed drug crimes in all three WPs (see Table 4 columns 8–10), the results show that the moderating effect of crime strategy significantly enhances the mediation effect of perceived crime benefit on future offending (β = 0.157, BCBCI = [0.021, 0.292]), criminal self-efficacy (β = 0.041, BCBCI = [0.005, 0.077]) and, ultimately, arrest avoidance (β = 0.027, BCBCI = [0.009, 0.045]). In contrast, for less-experienced drug dealers who committed drug crimes only in one or two WPs (see Table 4 columns 5–7), the results show that the moderation effect fails to make a significant alteration on the mediation effect of the perceived crime benefit on future offending (β = 0.036, BCBCI = [−0.029, 0.101]), criminal self-efficacy (β = 0.022, BCBCI = [−0.001, 0.054]) or, ultimately, arrest avoidance (β = 0.009, BCBCI = [−0.005, 0.023]) with the bootstrapping intervals containing zero. Since there is statistical evidence of a significant moderation effect in the group of experienced drug dealers but not in the group of less-experienced drug dealers, Hypothesis 3 is supported.

## 4. Discussion

The first hypothesis in this study predicted that the perceived crime benefits are associated with proactive criminal thinking and future offending when a series of variables are controlled. In addition, it predicted that the effect size of the mediation pathway varies when considering the use of a crime strategy. The second hypothesis that was tested in this study predicted that proactive criminal thinking mediates the relationship between the perceived crime benefit, criminal self-efficacy, and arrest avoidance when a series of variables are controlled. It was expected that the effect size of the mediation pathway in the second hypothesis would vary with different levels of crime strategy. Grouping the drug dealer sample into experienced and less-experienced drug dealers, the third hypothesis in this study predicted a discrepancy in the moderated mediation across the two subsamples. The statistical outcomes are consistent with the expectations of the three hypotheses.

The first main finding is the validation of the two mediation pathways. Specifically, criminal thinking is a factor in channeling the effect of the perceived crime benefit on future offending and efficacy in crime. From previous research [27], there appears to be a growing body of evidence supporting the proposition that perceived crime benefit could represent part of criminal thinking and lead to future offending or re-offending. The old adage that one of the best theories for predicting future behavior is the rational choice theory (evaluation of reward and cost) [43] may be accurate but is not, in and of itself, sufficient in guiding correction/intervention work among offenders. Cognitive mediation provides a pathway connecting the perceived benefit of crime with the perceived self-capacity in carrying out crime. The perceived crime reward shapes one’s criminal thinking, and it could further influence one’s expectation of dodging enforcement and one’s propensity to engage in future criminality. The short-term or impulsive perceived crime benefit plays a role in this process [27]. Other than that, proactive criminal thinking, fostered by a pre-planned behavioral approach, is believed to facilitate crime continuity through the pre-judgment of outcomes and the assessment of self-efficacy [10]. This is compatible with the idea of restrictive deterrence, which involves a dynamic evaluation of crime benefits and costs under considerations of a crime strategy. Proactive criminal thinking and restrictive deterrence are probably both a manifestation of thoughtfully reflective decision-making (TRDM) (see [74]) in the offender population.

Based on the statistical fit for the moderated mediation, the second finding is that crime strategy plays a role in reinforcing the positive relationship between the perceived crime benefit and proactive criminal thinking. Beyond that, the reinforcing effect can continue to play a role in subsequent influencing pathways. This finding echoes the idea that the perceived risk can exert an effect across several crime types. As Anwar and Loughran demonstrated, risk updating is not strictly based on the type of crime that an offender was charged with (i.e., it is not strictly crime-specific) [59]. Although the current study used a different model of argument from that of Anwar and Loughran [59] (they tested two crimes, an aggressive crime and an income crime), it also testifies to the point that drug dealers’ perceptions and cognitions of crime are partly associated with the manner in which they conduct property crime.

In addition to supporting the non-crime-specific feature of the crime strategy effect, the current finding contributes to quantifying the pathway through which restrictive deterrence can play a role. When interacting with the perceived crime benefit, a crime strategy is associated with proactive criminal thinking, criminal self-efficacy, and arrest avoidance. Jacobs and Cherbonneau demonstrated that using a crime strategy, accompanied by the capacity to recognize and adapt to risk, decreases one’s crime risk or at least one’s perceived crime risk [50]. Brezina and Topalli presented the interaction of crime strategy and crime benefit more directly but from a competitive perspective [54]. They built an empirical study exploring criminal self-efficacy and added crime strategy after illegal income as predictors in a hierarchical linear regression model. The authors found that the effect of crime strategy on self-efficacy completely masked that of illegal income. The competitive interaction between crime strategy and crime benefit was, presumably, derived from the fact that Brezina and Topalli [54] used a sample of property-based offenders, which led to a significant overlap between the two factors in the data structure. It has also been found that crime strategy is a valid predictor for crime income [51]. By contrast, the current paper investigated the effect of heterogeneous crime strategy using the sample of drug dealers and their use of a property crime strategy. Therefore, the overlap between the crime strategy and the crime benefit can be assumed to be smaller than the overlap in Brezina and Topalli’s study [54]. Regardless of the overlap, the previous findings suggested an association between crime strategy and the crime benefit. The current study further tested how the interaction of the two factors can impact crime perceptions. This allows us to take a fresh look at the relationship between crime strategy and crime benefit. Crime strategy and crime benefit should not necessarily be seen as only being in an adversarial relationship (the influence of one being overshadowed by the other) but can also be seen as being in a cooperative relationship (where the interaction terms that are generated by the two impact crime perception and cognition).

The current study also found different outcomes for the mediation and moderated mediation when dividing drug dealers based on their crime experience. The statistical outcome illustrates that the experienced drug dealers’ perceptions of arrest avoidance and intention to re-offend are directly or indirectly affected by the predictors. However, this is not the case for the less-experienced drug dealers. It is commonly accepted that an experience of arrest leads to an upward shift in perceived crime risk, and an experience of arrest avoidance leads to the reverse trend [45]. However, there is a difference between experienced and less-experienced offenders in how to evaluate and process those experiences. Experienced offenders presumably possess more expertise and have a high sensitivity to subtle clues in the crime environment, which will be automatically processed with less cognitive effort [55]. In turn, less-experienced offenders show less sensitivity in recognizing and integrating heterogeneous crime clues. In addition to a deficiency in sensitivity, less-experienced offenders are also “short-sighted” in updating their risk perceptions. Stafford and Warr found that less-experienced offenders, when compared to experienced offenders, put more weight on their current experience with apprehension when adjusting their risk perception, which means that less-experienced individuals pay more attention to a punishment that occurred recently [45]. In other words, less-experienced offenders put greater value on a punishment if it is “closer” to them in terms of time. This could explain the discrepancies in the outcomes of the mediation and the moderated mediation. On the one hand, the experienced drug dealers integrate details of the crime and related knowledge that is gained from other sources, and this influences their overall perception of crime. On the other hand, less-experienced drug traffickers focus more on the experience of committing a crime of the current crime type and less on integrating cross-crime guidance into their overall crime perception. The discrepancy outcome between experienced drug dealers and the less-experienced group allows an extension of the definition of “closer.” In addition to referring to temporal distance, the distance between crime types could also be considered in future research (see [66]).

Besides the primary outcome that was led by the three hypotheses, two statistical outcomes are worthy of further elaboration. One is that proactive criminal thinking played a more critical role in connecting the association between the perceived crime benefit and future offending (full mediation) than criminal self-efficacy and arrest avoidance (partial mediation). There are two potential explanations for the smaller proportion of indirect effect sizes for the second mediation pathway. One is that there are two mediators in the second pathway, but only one mediator in the first pathway. The second is that it is generally accepted that the salient role of proactive criminal thinking in future offending while its impact on crime capacity has been less well explored.

### Limitation

The current study has its limitations, one of which is the way in which proactive criminal thinking was measured. The quality of the measurement of proactive criminal thinking may affect the results of the outcome of the mediation and moderated mediation. The current paper used data from the Second RAND Inmate Survey. The questions that were related to criminal cognition in this survey were not identical to the one that was constructed by Walters [10], but some of the items can be theoretically mapped to sub-dimensions of proactive criminal thinking, such as mollification, entitlement and power orientation, and super-optimism. The use of items from large databases to form an ad hoc scale that correlates with criminal thinking has been seen in previous studies. In using an extensive database, Walters, in the absence of a very formal scale of criminal thinking, also used existing items in the database to form a scale that was similar to criminal thinking in his research [14,27,75]. Hence, to measure criminal thinking more effectively, there may need to be additional or different items.

The second limitation is the small size of the mediation effect and the moderated mediation effect. In this case, though, small does not equal insignificant. Mediation analyses are often characterized by small effects because of the multivariate nature of the analysis, low power [76], and the requirement for high control [77]. These problems result in low power, and low power is usually concomitant with mediation analysis. In addition, the present results are no exception to the rule that mediation effects are nearly always small [2,76]. Although power constraints and variable control requirements of mediation analysis present difficulties, the consistency of the outcomes indicate that the results have important implications for research and practice in criminal thinking and restrictive deterrence. In addition, the different mediation outcomes that were led by different levels of moderators is a sign of a causal relationship in the mediation pathways (see [78,79,80]).

The third limitation is the data bias that is caused by lack of timeliness and a lack of female drug offenders. The Second RAND Inmate Survey was conducted over 44 years ago, during which the environment and actors that were involved in drug market have changed somewhat. For example, drug business that was run by Chinese triads would result in a gang member taking into consideration collective interests rather than just individual interests [81], indicating individual characteristics compete with organizational ones. Similarly, female characteristics compete with male ones in drug trafficking [82]. Female drug offenders, often deemed as marginalized and passive victims in drug crime, are playing a more visible and less-subordinate role in drug crime [83], signifying that women’s proactive cognition and behavior towards to drug trafficking and trade deserve to be noticed by scholars and others. It can be expected that such data bias could deprive its findings of some reference, mainly at a practical level, i.e., formulating interventions in general. Nevertheless, the drug market is changing rapidly, and drug decriminalization and the marijuana legalization movement may lead to a fast expansion of the drug market, as shown in a new polling that was reported in August 2022 that marijuana use reached a record high as, for the first time, more Americans said they smoke marijuana than reported smoking cigarettes in the last week [84]. It may lead to an upsurge in the individual characteristics of criminal activity, such as the opportunity for women to develop their drug economy apart from men [83]. Meanwhile, the drug market is stable in terms of the transaction process. Low-level drug dealers, who hold specific and nuanced criminal strategies to complete the “last mile” of drug distribution, were frequently treated as research “entrance” for scholars focusing on the cognition of crime. In addition, the exploration of integrating restrictive deterrence within social learning and the social cognitive theoretical framework can be satisfied by the RAND data.

## 5. Conclusions

Restrictive deterrence was an important concept that emphasized criminal competence and crime strategy at the time that Walters introduced the dichotomy of proactive and reactive criminal thinking into criminology and criminal justice. As a next step toward developing a comprehensive concept of restrictive deterrence, quantifying the research findings from numerous qualitative studies could be expected. A specific approach would combine the cognitive skills of criminal strategy with models of criminal thinking. Restrictive deterrence research has long been dominated by a single-purpose claim, and limited by feasible quantitative research methods. It is time to integrate the various qualitative findings that have evolved into a more comprehensive and cohesive model. Mediation and moderation in statistics [85] allow scholars to investigate the interaction between the variables that are currently used to explain crime. It can be of assistance in integrating crime practice into a broader theoretical map. Getting on board with social learning theory and social cognitive theory is a meaningful attempt.

## Figures and Tables

**Figure 1 ijerph-19-11636-f001:**
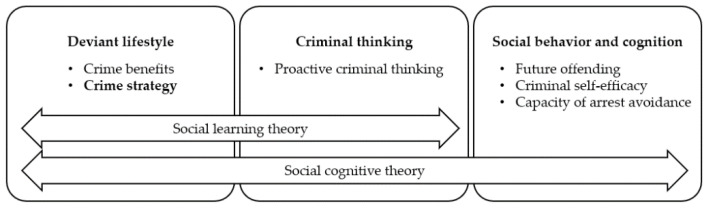
Embedding restrictive deterrence in framework of social learning theory and social cognitive theory.

**Figure 2 ijerph-19-11636-f002:**
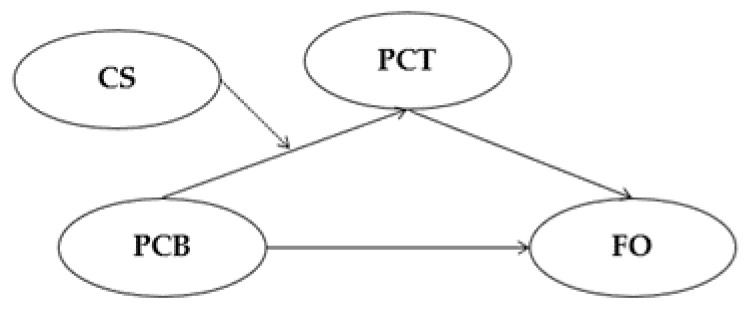
The mediation pathway of Hypothesis 1. Note. PCB = Perceived crime benefit, PCT = Proactive criminal thinking, FO = Future offending, CS = Crime strategy. The bold lines show mediation pathways (Hypothesis 1a), and the dashed line a moderation pathway (Hypothesis 1b).

**Figure 3 ijerph-19-11636-f003:**
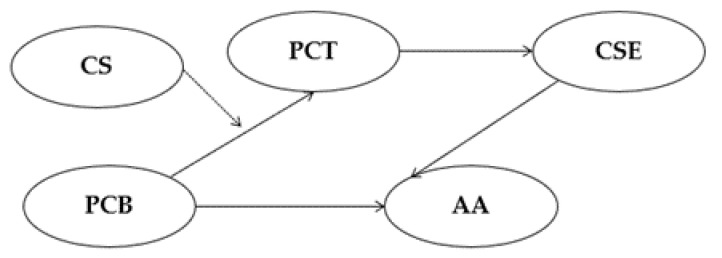
The mediation pathway of Hypothesis 2. Note. PCB = Perceived crime benefit, PCT = Proactive criminal thinking, CSE = Criminal self-efficacy, AA = Arrest avoidance, CS = Crime strategy. The bold lines refer to mediation pathways (Hypothesis 2a), and the dashed line to a moderation pathway (Hypothesis 2b).

**Figure 4 ijerph-19-11636-f004:**
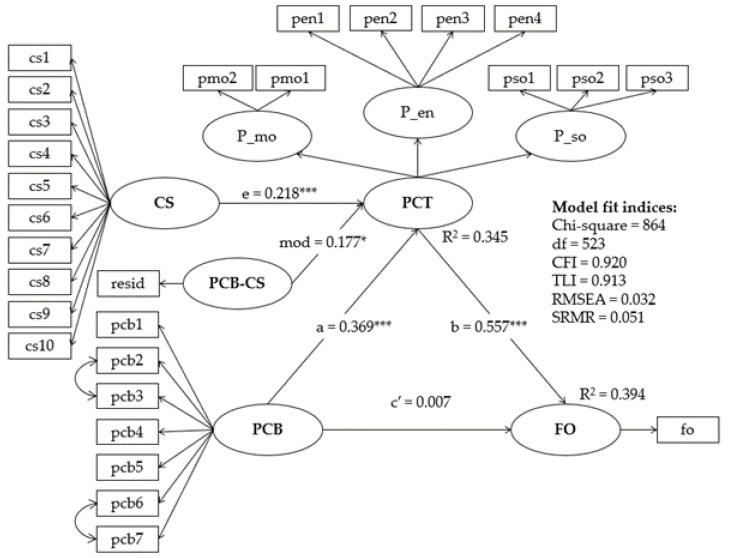
Standardized path coefficients of moderated mediation structural equation model (outcome variable = FO). * refers *p* < 0.01 and *** refers *p* < 0.001. Note. PCB = Perceived crime benefit, PCT = Proactive criminal thinking, P_mo = Mollification, P_en = Entitlement, P_so = Super optimism, FO = Future offending, CS = Crime strategy. PCB_CS = Interaction between PCB and CS.

**Figure 5 ijerph-19-11636-f005:**
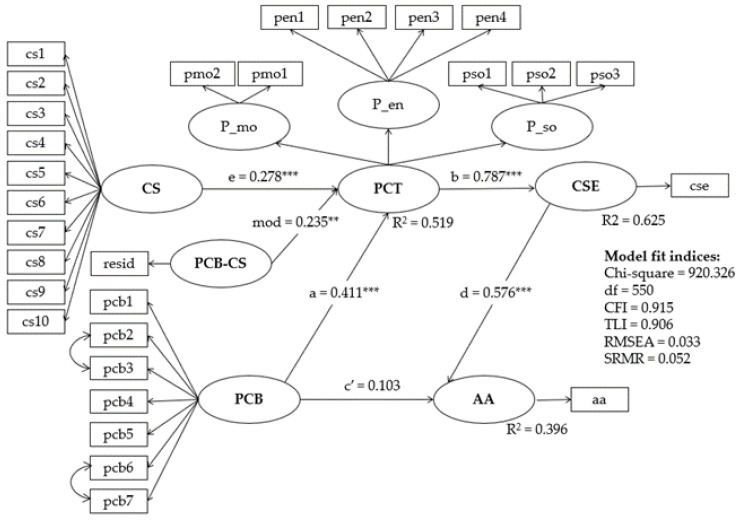
Standardized path coefficients of moderated mediation structural equation model (outcome variable = AA). ** refers *p* < 0.05 and *** refers *p* < 0.001. Note. PCB = Perceived crime benefit, PCT = Proactive criminal thinking, P_mo = Mollification, P_en = Entitlement, P_so = Super-optimism, CSE = Criminal self-efficacy, AA = Arrest avoidance CS = Crime strategy. PCB_CS = Interaction between PCB and CS.

**Table 1 ijerph-19-11636-t001:** Demographic statistics for drug dealer sample in the Second RAND Inmate Survey.

Variable	N. Valid	%/ Mean (Std. Dev, Min–Max)
Age	832	25.448 (6.64, 14–60)
Race	829	
Asian	4	0.50%
Black	310	37.40%
Chicano/Latino	95	11.50%
Indian/Native American	9	1.10%
White	395	47.60%
Other	16	1.90%
Education	834	
No schooling	3	0.40%
6th grade or less	22	2.60%
7th–9th grade	121	14.50%
10th–11th grade	298	35.70%
High school grade	165	19.80%
Some college	206	24.70%
College graduate	15	1.80%
Post-graduate study	4	0.50%
Marriage	826	
Married	147	17.80%
Widowed	12	1.50%
Divorced	116	14%
Separated	60	7.30%
Never married	491	59.40%
Experience in drug dealing *	850	
Committed drug dealing in WP3 only	266	31.30%
Committed drug dealing in WP3 and WP2 or WP1	219	25.80%
Committed drug dealing in WP3, WP2, and WP1	365	42.90%
Frequency of drug dealing	850	2.326 (1.76, 0–4)
Self-identity as a drug dealer	831	
No	432	52%
Yes	399	48%
Current incarceration for drug dealing	820	
No	705	86%
Yes	115	14%
No. of arrests for drug dealing	800	0.489 (1, 0–8)
No. of crime types	850	2.951 (2.24, 0–9)
No. of property crimes	850	2.392 (1.85, 0–7)
No. of violent crimes	850	0.559 (0.68, 0–2)
Length of WP3 (months)	841	14.432 (5.57, 1–25)

Note. * WP1 = window period 1; WP2 = window period 2; WP3 = window period 3.

**Table 2 ijerph-19-11636-t002:** Model fit indices and covariance of measurement model and structure model.

	Model Fit Indices						Covariance				
CFA model	Chi-square	df	CFI	TLI	RMSEA	SRMR	PCB	CS	FO	CSE	AA
Structure model	617.352	360	0.948	0.941	0.034	0.04					
PPCT	42.177	24	0.965	0.947	0.034	0.04	0.389 ***	0.302 ***	0.549 ***	0.447 ***	0.307 ***
PCB	52.478	12	0.96	0.93	0.08	0.046		0.258 ***	0.202 ***	0.358 ***	0.268 ***
CS	133.974	35	0.955	0.942	0.078	0.039			0.193 ***	0.265 ***	0.301 ***
FO										0.194 ***	0.336 ***
CSE											0.367 ***

Note. PCB = Perceived crime benefit, PCT = Proactive criminal thinking, CSE = Criminal self-efficacy, AA = Arrest avoidance. *** *p* < 0.001.

**Table 3 ijerph-19-11636-t003:** Mediation analysis of perceived crime benefit as predictor of future offending, criminal self-efficacy, and arrest avoidance via proactive criminal thinking.

Pathways	BCBCI			Type of Mediation
	β	Lower	Upper	
**FO (outcome measure)**				Full mediation
PCB → FO (Direct effect)	−0.108	−0.339	0.555	
PCB → PCT → FO (Indirect effect)	0.583	0.252	0.914	
**CSE (outcome measure)**				Partial mediation
PCB → CSE (Direct effect)	0.242	0.109	0.375	
PCB → PCT → CSE (Indirect effect)	0.108	0.024	0.193	
**AA (outcome measure)**				Partial mediation
PCB → AA (Direct effect)	0.116	0.05	0.182	
PCB → PCT → CSE → AA (Indirect effect)	0.044	0.013	0.075	

Note. PCB = Perceived crime benefit, PCT = Proactive criminal thinking, FO = Future offending, CSE = Criminal self-efficacy, AA = Arrest avoidance, CS = Crime strategy.

**Table 4 ijerph-19-11636-t004:** Moderated mediation analysis of perceived crime benefit as a predictor of future offending, criminal self-efficacy, and arrest avoidance via proactive criminal thinking when moderated by crime strategy.

Pathways	Drug Dealer (*n* = 850)			Less-Experienced Group (*n* = 485)			Experienced Group (*n* = 365)		
	BCBCI			BCBCI			BCBCI		
	β	Lower	Upper	β	Lower	Upper	β	Lower	Upper
**PCT (outcome measure)**									
PCB (predictor)	0.161	0.076	0.246	0.1	0.009	0.192	0.24	0.096	0.385
CS (moderator)	0.061	0.026	0.097	0.045	0.003	0.087	0.073	0.015	0.132
PCB_CS (interactor) *	0.017	0.002	0.032	0.008	−0.007	0.002	0.031	0.005	0.056
**PCB** → **PCT** → **FO**									
mean(CS) − 1SD	0.754	0.335	1.173	0.460	0.088	0.833	1.246	0.185	2.307
mean(CS)	0.814	0.380	1.248	0.486	0.104	0.868	1.368	0.251	2.486
mean(CS) + 1SD	0.874	0.420	1.328	0.512	0.115	0.909	1.490	0.309	2.671
Index of moderated mediation	0.079	0.012	0.147	0.036	−0.029	0.101	0.157	0.021	0.292
**PCB** → **PCT** → **CSE**									
mean(CS) − 1SD	0.023	0.084	0.363	0.213	0.013	0.413	0.251	0.024	0.477
mean(CS)	0.246	0.099	0.392	0.229	0.023	0.435	0.283	0.037	0.529
mean(CS) + 1SD	0.268	0.113	0.423	0.224	0.031	0.458	0.315	0.048	0.582
Index of moderated mediation	0.03	0.007	0.052	0.022	−0.001	0.054	0.041	0.005	0.077
**PCB** → **PCT** → **CSE → AA**									
mean(CS) − 1SD	0.118	0.058	0.177	0.106	0.028	0.185	0.161	0.049	0.272
mean(CS)	0.129	0.067	0..191	0.113	0.033	0.192	0.182	0.063	0.302
mean(CS) + 1SD	0.140	0.075	0.205	0.119	0.037	0.2	0.204	0.075	0.333
Index of moderated mediation	0.015	0.005	0.025	0.009	−0.005	0.023	0.027	0.009	0.045

Note. PCB = Perceived crime benefit, PCT = Criminal thinking, FO = Future offending, CSE = Criminal self-efficacy, AA = Arrest avoidance, CS = Crime strategy. * PCB_CS = Interaction between PCB and CS. PCB_CS is formed in three steps: (1) product term equals to the sum score of 7 items in PCB (pcb) multiply mean score of 10 items in CS (cs); (2) product term is regressed on pcb and cs, and residual formed is kept; (3) residual (resid) is used as observe variable to form PCB_CS as a single-item factor. The procedure of forming PCB_CS is a simplified version of the residual centering approach of Little et al. [73].

## Data Availability

Publicly available datasets were analyzed in this study. This data can be found here: https://www.icpsr.umich.edu/web/NACJD/studies/8169/versions/V2 (accessed on 1 July 2022). The R code for the analysis can be found at GitHub: https://github.com/guanxin644/Analysis-of-ICSPR-08169-RAND-Second-Inmate-Survey (accessed on 1 July 2022).

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
