# Peer review of "Proactive Criminal Thinking and Restrictive Deterrence: A Pathway to Future Offending and Sanction Avoidance"

_ijerph, 2022, doi:10.3390/ijerph191811636_

Round 1
Reviewer 1 Report
Table 2 can be improved if the most relevant significant results are clearly indicated.
Author Response
Q: Table 2 can be improved if the most relevant significant results are clearly indicated.
- R: Thanks for pointing this out. We agree with the comment. However, we adopted SEM as the main analytic strategy in this revision, Table 2 is converted completely and presented the statistic of model fit indices and covariance of measurement model and structure model. We still follow this comment that only model fit and covariance are kept (Line 420-421). Meanwhile, we also simplified the Table 1 (Line 394-395) which only contains mean, std.dev, min, max and percentage.

Reviewer 2 Report
Theoretical Issues
• The bibliographic review is abundant and up-to-date, although it has unjustified biases that I suggest should be corrected. 27% of the references in the Introduction correspond to G. D. Walters, who is explicitly indicated as a theoretical reference, however, this author in his studies is based on Albert Bandura's Theory of Social Learning, who is not cited in the Article. This is important, since the concepts of Self-Efficacy and Vicarious Learning were formulated and developed by Bandura and the concept of Criminal Thinking is a derivation of Moral Disengagement, also formulated by this author as an explanation of social deviation. It is of scientific rigor to establish the original line of thought that gives rise to the formulation of the problem.
• In the article there is a clear definition of Criminal Thinking (1.1, first paragraph), but this is not the case of Restrictive Deterrence, which is exposed extensively without an explicit definition. This is very necessary to specify whether the consequences of the crime (economic and legal) are being emphasized in a positive and negative sense (gain/loss) or self-efficacy as an anticipation of success, or both.
• In the conceptualization of Criminal Thinking, and following Walters, a proactive form is distinguished from a reactive one and the relationship of each one with the Restrictive Deterrence is theorized, however, this distinction is not incorporated in the design, being fundamental its demonstration (by reviewing the content of the 10 Thinking items, the two forms indicated could theoretically be distinguished).
Methodological Issues
• The study sample has characteristics that must be explained in the method or, failing that, in the limitations of the study, since they affect the analyses, the results and the conclusions, and may be considered biases of the study.
· The data is too old (44 years) and corresponds to a different reality from that of the authors. This means that the data comes from a social and criminological context that is very different from the current one in which the study will be published, with the change experienced by the drug trafficking criminal organization in America being especially relevant, where individual characteristics lose weight compared to organizational ones.
· The fact that they are only men should also be discussed, since in recent decades the rate of sanctioned women in America has increased, showing a very high prevalence of crimes associated with drug trafficking.
• It is not described how the experienced and less experienced subsamples were configured, it could be presumed that it corresponds to the presence of crimes in the different windows, but it is not indicated to which time periods WP1 and WP2 correspond either. It is very necessary to clarify this, establishing the comparison between both groups, since the observed effect could be mediated by age, since the sample ranges from 14 to 60 years with a mean of 24, and the effect found can be associated with the particular characteristics of juvenile delinquency.
• The dependent variables (future delinquency and avoidance of arrest) are well defined, but have an essentially different character from the other variables introduced in the model, since they correspond to future expectations and not to lived experiences (self-efficacy) or antisocial attitudes (thinking criminal), this together with the measurement scales used require a better justification for the analysis carried out.
• Although the analyzes carried out allow solving the research problem, I suggest adding the estimation of the model's goodness of fit using structural equation analysis, a procedure that provides greater certainty regarding mediation and moderation effects.
Final questions
• Due to the methodological characteristics of the study, I suggest that it be proposed as a theoretical investigation, which seeks to demonstrate the hypotheses proposed with secondary data.
Although the conclusion is logical, "...it could be helpful to explain individual differences in the practice of crime (...and...) shed light on possible ways of prevention, intervention and response.", it does not seem very pertinent to me given the data age. I suggest refocusing on theory validation.
Author Response
- Comments on theoretical issues
Q: The bibliographic review is abundant and up-to-date, although it has unjustified biases that I suggest should be corrected. 27% of the references in the Introduction correspond to G. D. Walters, who is explicitly indicated as a theoretical reference, however, this author in his studies is based on Albert Bandura's Theory of Social Learning, who is not cited in the Article. This is important, since the concepts of Self-Efficacy and Vicarious Learning were formulated and developed by Bandura and the concept of Criminal Thinking is a derivation of Moral Disengagement, also formulated by this author as an explanation of social deviation. It is of scientific rigor to establish the original line of thought that gives rise to the formulation of the problem.
- R: We thank the reviewer for raising the concern of lacking original line of theory background of criminal thinking. We agree with this comment. As suggested, the social learning theory and social cognitive theory of Bandura has been added during the introduction of criminal thinking (Line 42-50). In addition, the concept of moral disengagement proposed by Brandura is further used to distinguish the proactive and reactive criminal thinking (Line 52-66).
Q: In the article there is a clear definition of Criminal Thinking (1.1, first paragraph), but this is not the case of Restrictive Deterrence, which is exposed extensively without an explicit definition. This is very necessary to specify whether the consequences of the crime (economic and legal) are being emphasized in a positive and negative sense (gain/loss) or self-efficacy as an anticipation of success, or both.
- R: Thanks for pointing this out. We agree with this comment. As suggested, a more detailed description of restrictive deterrence is added, including its origin and definition, its performance in drug crime and related research in other types of crime (Line 107-123). Subsequently, we illustrated its power on leading crime behavior and crime cognition to echoes the kernel of restrictive deterrence -- dynamically assessing benefit/cost of crime and taking action to reverse an undesirable state or enhance a desirable one (Line 124-140).
Q: In the conceptualization of Criminal Thinking, and following Walters, a proactive form is distinguished from a reactive one and the relationship of each one with the Restrictive Deterrence is theorized, however, this distinction is not incorporated in the design, being fundamental its demonstration (by reviewing the content of the 10 Thinking items, the two forms indicated could theoretically be distinguished).
- R: Thanks for offering such advice to improve the construct of variables. We agree with this comment. In the revision, we put emphasize on proactive criminal thinking rather than both proactive and reactive parts because the kernel of restrictive deterrence echoes the pre-planned approach to crime which is the feature of proactive criminal thinking (Line 67-94). Three sub-dimensions of proactive criminal thinking, including mollification, entitlement and super optimism, are elaborated (Line 56-59; Line 76-80) to set a framework for constructing the measurement using items in the RAND. As the items in RAND are not totally identical to the Walters’ version, items in RAND that related to the psychology and perception on crime were theoretically mapped to each of three sub-dimensions of proactive criminal thinking to form measurement of mollification, entitlement and super optimism, and finally, the proactive criminal thinking (Line 282-296).
- Comments on methodological issues
Q: The study sample has characteristics that must be explained in the method or, failing that, in the limitations of the study, since they affect the analyses, the results and the conclusions, and may be considered biases of the study. (1) The data is too old (44 years) and corresponds to a different reality from that of the authors. This means that the data comes from a social and criminological context that is very different from the current one in which the study will be published, with the change experienced by the drug trafficking criminal organization in America being especially relevant, where individual characteristics lose weight compared to organizational ones. (2) The fact that they are only men should also be discussed, since in recent decades the rate of sanctioned women in America has increased, showing a very high prevalence of crimes associated with drug trafficking.
- R: Thanks for pointing this out. We agree with this comment. The nature of the RAND data (outdated and only male) might fall short of offering the most appropriate advice on intervention or policing work in current days, which we have made it as a research limitation (Line 621-632). Meanwhile, we also notice that competitions between individual characteristics competes vs. organizational characteristics, and female characteristics vs. male characteristics are not a zero-sum or absolute situation. Individual characteristics in drug crime may upsurge again under the trend of drug decriminalization in several countries (Line 632-636). Female characteristics share commons with their counterparts because the role they play in drug crime cultivate their cognition and perception on crime activity (Line 636-642). The nature the RAND data might not highly informative with in intervention orientation caused by data bias, it would not discount its value in theoretical investigation (Line 642-643).
Q: It is not described how the experienced and less experienced subsamples were configured, it could be presumed that it corresponds to the presence of crimes in the different windows, but it is not indicated to which time periods WP1 and WP2 correspond either. It is very necessary to clarify this, establishing the comparison between both groups, since the observed effect could be mediated by age, since the sample ranges from 14 to 60 years with a mean of 24, and the effect found can be associated with the particular characteristics of juvenile delinquency.
- R: Thanks for bringing this to our attention. We agree with this comment. We first test the association between age and experience in drug crime. T-test was performed, the results show that the mean_ageWP1 = 24.3 and mean_ageWP2 = 24.6 do not significantly differ from each other; however, mean_ageWP1 and mean_ageWP2 both differ from mean_ageWP3 = 26.7. Similarity, KS-test was also performed to examine the distribution commonness of participants’ age in group WP1, WP2 and WP3. The result report that distribution_ageWP1 is same as distribution_ageWP2 while both differ from distribution_ageWP3. It is indicated a difference of age mean value and distribution between people in WP1 and in WP3, and between WP2 and WP3. We further test the liner regression using the full sample: DV = experience in drug crime, IV = age, B = 0.03, p-value = 2.88e-06. Another liner regression using only sample WP1 and WP2: DV = experience in drug crime, IV = age, B = 0.001, p-value = 0.604.
The results of T-test, KS-test and liner regression show that the positive association with age and experience in drug crime, which is in line with the common sense. One thing to note is that the association is mainly results from the difference between WP3 and other two groups. Hence, we combined WP1 and WP2 as less-experienced group (Line 463-465), and WP3 be the experienced group (Line 458-460).
It is inevitably that experience associates with age given that both grow with time. The experience–age association is not just in criminal activity but in other areas as well. What is important is to clear in what approach they connect with each other. and design the analysis to avoid obtaining a pseudo-effect. We added age as a control variable in all SEM analysis (Supplementary Materials Table S3). The effects of key variables were still valid. Though, this is not the best way to isolate the age effect, the results are still convincible.
Q: The dependent variables (future delinquency and avoidance of arrest) are well defined but have an essentially different character from the other variables introduced in the model, since they correspond to future expectations and not to lived experiences (self-efficacy) or antisocial attitudes (thinking criminal), this together with the measurement scales used require a better justification for the analysis carried out.
- R: Thanks for bringing this up. We agree with this comment. As suggested, we adopted a theoretical investigation in this paper by embedding restrictive deterrence/crime practice into a broader framework of social learning theory and social cognitive theory by hooking with proactive criminal thinking (Line 214-218, Figure 1). Social learning theory emphasizes that reinforcement and assimilation of deviant behaviors, attitudes and skills in the company of deviants, which help to understand the formation of criminal thinking (Line 45-47). Social cognitive theory indicates the cognitive and behavioral changes resulted from criminal thinking, specifically, the mechanisms that can explain attributions of deviant activity, expectations of deviant outcome and perceptions of self-competence (Line 47-50).
The dependent variables (future delinquency and avoidance of arrest) indicate a propensity to commit crime in future and capacity to avoid the negative outcome, which are mainly accounted on previous performance in crime activity.
Q: Although the analyzes carried out allow solving the research problem, I suggest adding the estimation of the model's goodness of fit using structural equation analysis, a procedure that provides greater certainty regarding mediation and moderation effects.
- R: Thanks for offering such advice to improve the statistics analysis. We agree with this point. We conducted EFA (Line 397-403; Supplementary Materials Table S1), CFA (Line 404-422; Table 2; Supplementary Materials Table S2) and SEM (Line 423-491; Table 3; Table 4; Supplementary Table S3; Figure 4; Figure 5), using same data to test the mediation and moderated mediation model. The direction and significance level of results are same as previous version conducted by {PROCESS}. The slight difference is that items used to form perceived crime benefit (PCB), proactive criminal thinking (PCT), and crime strategy (CS) are not identical to the last manuscript because the standard of SEM more complex, which is mainly rest upon factor loading and model fit.
- Comments on theoretical investigation and intervention orientation
Q: Due to the methodological characteristics of the study, I suggest that it be proposed as a theoretical investigation, which seeks to demonstrate the hypotheses proposed with secondary data.
- R: Thanks for bringing up the concern. We agree with the comment. As suggested, we fit the model proposed in current study in frameworks of social learning theory and social cognitive theory (Line 214-218, Figure 1). We treat this paper as a meaningful attempt to integrating restrictive deterrence into a broader theoretical map, such as social learning theory and social cognitive theory (Line 656-657).
Q: Although the conclusion is logical, "...it could be helpful to explain individual differences in the practice of crime (...and...) shed light on possible ways of prevention, intervention and response.", it does not seem very pertinent to me given the data age. I suggest refocusing on theory validation.
- R: Thanks for pointing this out and referring to a feasible solution to us. We agree with this comment. The previous intervention orientation of this manuscript has been changed into a theoretical investigation by embedding restrictive deterrence/crime practice into a broader framework of social learning theory and social cognitive by hooking with proactive criminal thinking (Line 214-218, Figure 1). Meanwhile, the nature of the RAND data (outdated and only male) might fall short of offering the most appropriate advice on intervention or policing work in current days, which we have made it as a research limitation (Line 621-643).

Reviewer 3 Report
At the end of the day this paper would be of most interest to psychologists and correctional practitioners directly involved in rehabilitation. Further the interventions involved would demand a high level of sophistication and staff training. Unfortunately, correctional authorities have consistently demonstrated that they are unable to deliver on sophisticated interventions, particularly where offenders receive extremely short sentences.
Author Response
Q: At the end of the day this paper would be of most interest to psychologists and correctional practitioners directly involved in rehabilitation. Further the interventions involved would demand a high level of sophistication and staff training. Unfortunately, correctional authorities have consistently demonstrated that they are unable to deliver on sophisticated interventions, particularly where offenders receive extremely short sentences.
- R: Thanks for pointing this out. We agree with this comment. An effective intervention cannot be achieved without fine theoretical construction and rich practical experience. The nature of the RAND data (outdated and only male) and second-hand analysis based on it might fall short of offering the most appropriate advice on intervention or policing work in current days, which we have made it as a research limitation (Line 621-643). Alternatively, the previous intervention orientation of this manuscript has been changed into a theoretical investigation by embedding restrictive deterrence/crime practice into a broader framework of social learning theory and social cognitive theory by hooking with proactive criminal thinking (Line 214-218, Figure 1).

Round 2
Reviewer 2 Report
The comments and observations made previously were well received by the authors, introducing corrections that in my opinion have considerably improved the article in its current form, especially due to the reorientation as a theoretical article, based on secondary data.